# Clarifying the Differences between Patients with Organic Tics and Functional Tic-Like Behaviors

**DOI:** 10.3390/healthcare11101481

**Published:** 2023-05-19

**Authors:** Kaja Andersen, Ida Jensen, Kirstine Birkebæk Okkels, Liselotte Skov, Nanette Mol Debes

**Affiliations:** 1National Tourette Syndrome Clinic, Department of Pediatrics, Copenhagen University Hospital—Herlev and Gentofte, 2730 Herlev, Denmark; 2Department of Clinical Medicine, University of Copenhagen, 2200 Copenhagen, Denmark

**Keywords:** Functional Tic-Like Behaviors, Tourette Syndrome, comorbidities

## Abstract

Due to the global increase in the number of patients with Functional Tic-Like Behaviors (FTLB), it has become increasingly important to find reliable differences between this patient group and patients with organic tics (OTs), which can be used in differential diagnosis. The purpose of this retrospective study was to critically examine both established and suggested differences between the patient groups. A total of 53 FTLB patients and 200 OT patients were included. Several findings from the current literature were replicated in this study: Compared to patients with OTs, patients with FTLB had significantly more complex tics, were older at symptom onset, were more likely to be female, and were less likely to have family members with tics. Furthermore, the study also revealed new differences between the groups: Patients with FTLB had significantly more family members with a psychiatric disorder, were more likely to have experienced an adverse psychosocial event immediately before symptom onset, and had significantly fewer simple tics. Finally, this study was unable to replicate the previously found differences in comorbidities between patients with OTs and FTLB. These findings could contribute significantly to the understanding of FTLB’s etiology and to improve diagnosis, as including the presence of simple tics and comorbidities in the diagnostic criteria might be discussed in future studies.

## 1. Introduction

Functional Tic-Like Behaviors (FTLB) is a non-organic neurological disorder that belongs to the larger group of functional movement disorders (FMDs). FMDs are characterized by emulating the symptoms of a specific organic motor disorder such as dystonia or epilepsy [1]. FTLB mimic the tics seen in organic tic disorders, such as Tourette Syndrome (TS) and Chronic Tic Disorder (CTD) [2,3]. Being able to differentiate between FTLB and organic tics (OTs) in the clinical environment is important because they require different treatments and have different prognoses. FTLB respond inconsistently to both the behavioral and medical therapy used for OTs, and psychoeducation is instead recommended as a first-line treatment [4]. While other FMDs can be differentiated from their organic counterparts based on high patient distractibility and suppressibility, distinguishing FTLB from OTs may represent a distinct challenge as high trait distractibility and suppressibility are also present in organic tic disorders [5,6]. Instead, the literature has striven to differentiate FTLB from OTs by comparing the demographics and tic phenomenology of the two patient groups.

The demographics of patients with OTs are well established. OTs disproportionately affect boys, with on average four male patients to every female patient [7]. Over half of OT patients also fulfill the diagnostic criteria for one or more psychiatric comorbidities, the most frequent being Attention Deficit Hyperactivity Disorder (ADHD) and Obsessive-Compulsive Disorder (OCD) [8], however, Autistic Spectrum Disorder (ASD) is also common [9]. Furthermore, OTs tend to run in the family, showing high levels of heritability [10,11]. OTs develop gradually between the ages of 4 to 6 years old, and the patients are usually diagnosed at 8 to 9 years old [12,13,14]. In the younger age groups, the patients’ tics are predominantly simple motor tics, which involve only a single muscle group, and simple phonic tics, consisting of singular, nonsensical sounds. Complex tics, involving multiple muscle groups or sounds, usually develop as the patients grow older, but are seen in fewer patients compared to simple tics, and patients tend to have fewer complex tics than simple tics [15,16]. Debilitating behaviors, such as self-injurious behaviors or tics that are socially inappropriate (coprophenomena) are relatively rare in the population [17,18]. 

Comparatively less is known about patients with FTLB, mostly due to the relative recency of academic interest in the subject. While the first case reports describing FTLB were published at least as early as 1969 [19], and two reports were published in the 1990s [20,21] the disorder only began to receive consistent attention in the mid-2010s [22,23]. In 2014, Baizabal-Carvallo and Jankovic examined nine adult patients with tics-like symptoms [22]. Compared to 273 patients with established TS, the patients with FTLB were significantly older at symptom onset and more frequently female. Furthermore, the FTLB patients had no family history of tics and experienced only mild to no reduction of their tics when medicated. These patient characteristics have been corroborated in later studies. Demartini and colleagues (2015) examined 11 adult patients with FTLB. Core characteristics of the FTLB group were found to be a high frequency of female patients, late tic onset, high tic severity at onset, lack of family history of tics, high rates of anxiety and depression as comorbidities, and the presence of a trigger, such as an acute illness or psychosocial stress immediately before symptom onset [6]. These results set FTLB patients apart from patients with OTs, who, as highlighted above, are predominantly male and normally have a childhood tic onset, a positive family history of tics, high rates of ADHD and OCD as comorbidities, and no notable triggers before symptom onset [2]. 

In 2020, a sudden increase in the number of pediatric patients with unconventional tics, was observed in medical clinics worldwide. These tics, dubbed ‘TikTok tics’ or “mass social media-induced tics” due to their association with the use of the social media platform TikTok, were quickly determined to be a subtype of FTLB [5,24,25,26]. In a series of case studies, Hull and Parnes (2021) described six teenage girls between the ages of 13–16 years old with an abrupt onset of tics of high severity and complexity and with a high prevalence of self-injurious behaviors [27]. In Paulus and colleagues (2021), 13 pediatric patients with FTLB, whom all reported that they had watched tics-related content on TikTok, were compared to 13 age and sex-matched TS patients. It was found that the patients with FTLB had a significantly later symptom onset and were significantly more likely to have predominantly complex motor tics and an abrupt symptom onset, while family history and comorbidities did not differ between the groups [28]. Pringsheim and colleagues (2021) examined 20 pediatric patients with FTLB and compared them to a clinic population with OTs. Compared to the OT patient group, the FTLB patients were again significantly older both at onset and at first clinic visit, significantly more likely to be female and to have major depressive disorder and anxiety disorders and had a higher tic severity. The FTLB patients were generally characterized by abrupt onset of symptoms with a high number of complex tics but a smaller number of simple tics. Furthermore, multiple patients experienced self-injurious behaviors and coprophenomena [29]. Han and colleagues (2022) found the same differences in demography and comorbidities when comparing 22 pediatric FTLB patients with 168 OT patients. Furthermore, compared to the patients with OTs, significantly more of the FTLB patients experienced self-injurious behaviors, coprophenomena, hospitalizations, and school absences [30]. Finally, Trau and colleagues (2022) examined 198 patients who, after an initial analysis, were divided into three groups: organic tic disorder, functional tic disorder, and mixed tic disorder, with the latter category including children with both OTs and FTLBs. Compared to the patients with OTs significantly more of the patients with functional or mixed tics were female, suffered from self-injurious behaviors and coprophenomena, and had anxiety or OCD. Furthermore, the patients with functional and mixed tics had significantly higher tic severity scores, while fewer of them had a family history of tics [31]. Thus, like their adult counterparts, pediatric patients with FTLB differ from pediatric OT patients in terms of sex, age at onset, comorbidities, family history, and tic severity. Based on these differences, two checklists of diagnostic criteria for FTLB have been proposed [31,32]. Both checklists include the criteria of age 12 or older at onset, abrupt tic onset, and at least some complex tics and/or tic behaviors (e.g., long bursts of tics, blocking tics, coprophenomena, and self-injurious behaviors) as mandatory requirements for an FTLB diagnosis. Furthermore, both checklists highlight the presence of anxiety and depression as comorbidities as a minor criterion, while Trau and colleagues also include a lack of family history of tics [31,32]. 

The literature comparing pediatric patients with FTLB and OTs is a growing field, but it is currently limited by several different problems, two of which are small sample sizes and the presence of confounding variables. The small sample sizes are a constraint created by the small patient group; the resulting impairment in generalizability can be ameliorated by replication from different research groups. As for the second problem, confounding variables such as age and sex are known to influence both tic expression and the presence of comorbidities [7,33,34]. Therefore, it is important to correct for age and sex in analyses of tic and comorbidity differences between FTLB and OT patients to exclude the possibility that the sex and age differences between the groups can account for the results. Such a correction has to the best of our knowledge not been employed in any studies except for Paulus and colleagues and Pringsheim and colleagues [28,29]. 

Thus, the purpose of this study is to replicate the already established differences between pediatric patients with FTLB and OTs to achieve greater generalizability and test whether these differences persist after correcting for age and sex. This was performed by examining the demography and tics of one of the largest groups of pediatric patients with FTLB to date and contrasting these with pediatric patients with OTs. Finally, these results will be related to the proposed diagnostic criteria in the discussion. 

## 2. Materials and Methods

This retrospective study was approved by the Danish data protection authority (R-22052711). 

### 2.1. Population

The FTLB sample consisted of all FTLB patients who attended their first visit to The National Tourette Syndrome Clinic between May 2020 and April 2022. Despite FTLB being a recognized phenomenon previously, May 2020 was decided on as starting point as this is when this group started to be consistently categorized at our clinic. 

At that time, FTLB did not yet have an accepted list of diagnostic criteria, and as such, the diagnosis was made based on the assessment of senior child neurologists with many years of experience in diagnosing tic disorders. If there was any doubt about the patient’s diagnosis, clinical consensus was reached via a multidisciplinary conference. The first 23 of the FTLB patients have been described in a previous study [35]. 

The OT sample consisted of all OT patients who attended their first visit to the National Tourette Syndrome Clinic between May 2020 and July 2021. To be included, the OT patients had to fulfill the Diagnostic and Statistical Manual V criteria for Tourette Syndrome or Chronic Tic Disorder as determined by experienced senior child neurologists. The only exclusion criterion was FTLB as a suspected comorbidity which was true for 14 of the OT patients. 

### 2.2. Outcome Measures

All included patients’ medical records were thoroughly examined for information pertaining to psychiatric comorbidity, family history, possible symptom triggers, and tic phenomenology. This information was initially collected through a standard systematic medical interview which is routinely conducted by the attending doctors when a new patient is first seen as a part of the diagnostic process. This interview includes questions about all the measures described below. 

#### 2.2.1. Psychiatric Comorbidity

Psychiatric comorbidity was noted if the patient’s medical record contained diagnostic records from the child and adolescent psychiatry unit or if the patient and/or their guardian(s) stated a diagnosis during the systematic medical interview. Only diagnoses made before the referral to the clinic were included. The diagnoses were recorded using a yes/no approach to each diagnosis in the dataset.

#### 2.2.2. Family History

Family history was recorded based on the information provided by patients and their guardian(s) during the systematic medical interview. The presence of psychiatric diagnoses in first-degree relatives was registered using a yes/no approach to each diagnosis in the dataset. 

#### 2.2.3. Potential Triggers

During the systematic medical interview, all patients, regardless of the suspected diagnosis, were asked whether they had experienced any deterioration in their physical, mental, or social well-being leading up to the tic onset. If present, these precipitating events were described in detail in the medical record but were recorded in the dataset using a yes/no approach due to statistical reasons. 

#### 2.2.4. Type and Number of Tics

For the FTLB patients, information about the number and specific appearance of current tics was collected through the systematic medical interview and noted in the medical records. The resulting descriptions were used to record both the presence and total amount of simple and complex vocal and motor tics in the dataset. For the OT patients, this data was instead derived from their YGTSS score [36]. The YGTSS score was not routinely collected for the FTLB patients. Furthermore, the presence of four distinct behaviors was specifically recorded as they appear as individual items in the YGGTS: self-injurious behaviors, long bursts of tics, unrestrained speech, and coprophenomena. Self-injurious behaviors were defined as behaviors that caused the patient pain, long bursts of tics as periods of at least 30 s of constant ticking, unrestrained speech as the patient uttering several sentences uncontrollably, and coprophenomena as tics which included words or actions which are socially unacceptable. The presence of each tic type was registered and included in the total number of tics. In instances of doubt about the categorization of a tic, a senior child neurologist was consulted. 

### 2.3. Statistics

To examine whether the recorded measurements were related to patient diagnosis (FTLB or OT), several different types of analyses were used. Depending on the nature of the outcome variable and the presence of confounders, either chi-square tests, Mann–Whitney U, binary logistic regression, or negative binomial regression was used (Table 1). All analyses were performed in SPSS version 28.

## 3. Results

### 3.1. Cohort Characteristics

A total of 53 FTLB patients were included in the analysis. Of these patients, 50 were female and 3 were male. The mean age at symptom onset was 13.7 years (SD = 2.4, range = 6–19.8), and the mean age at the first visit was 14.9 years (SD = 2.0, range = 11–20.2 years old). Of the FTLB sample, 30 patients confirmed exposure to tics on social media. These patients had significantly higher odds of self-injurious behaviors (OR = 5.061, 95% CI [1.37, 18.72], *p* = 0.015) but did otherwise not differ from the patients who had not consumed tics-centered social media.

A total of 200 OT patients were also included. Of these, 139 were male, and 61 were female. The mean age at symptom onset was 6.4 years (SD = 2.5, range = 1.5–14 years old), and the mean age at the first visit was 10.4 years (SD = 2.9, range = 2.9–17.9 years old). The YGTSS score, including the number and severity of the tics was available for 87 of the OT patients. There were no significant differences in age, sex, and family history between these 87 patients and the rest of the OT sample.

### 3.2. Descriptive Statistics

Compared to the 200 patients with OTs, a higher percentage of the patients with FTLB had psychiatric comorbidities (Table 2). In particular, anxiety, ASD, OCD, and other psychiatric diagnoses were more frequent among patients with FTLB. A third of the OT sample had first-degree relatives with tics (Table 3). This was only the case for 11.3% of patients with FTLB. Conversely, a higher percentage of patients with FTLB had first-degree relatives with anxiety, ASD, and other psychiatric diagnoses. Furthermore, a higher percentage of patients with FTLB had experienced a psychosocial trigger immediately before their symptom onset (Table 3). The most frequently occurring triggers and precipitating events were consequences of the COVID-19 pandemic, exams, bullying, disease, or conflicts in the family. Of the 87 OT patients whose tic number and type were available, almost all patients had simple motor tics (Table 4), with a median number of four different simple motor tics in total (Table 5). Simple motor tics were seen in a comparatively lower percentage of the FTLB patients, who only had a median of one tic of this type. The opposite pattern was seen for complex motor and vocal tics, observed in a much higher percentage of patients with FTLB and in higher numbers compared to patients with OTs. Finally, a higher percentage of patients with FTLB had self-injurious behaviors, long bursts of tics, unrestrained speech, and coprophenomena (Table 4). 

### 3.3. Statistical Tests

#### 3.3.1. Age and Sex

There were significant differences between patients with FTLB and patients with OTs in terms of age and sex. Of the patients with FTLB, significantly more were female (χ^2^(1) = 69.342, *p* < 0.001). Furthermore, the patients with FTLB were significantly older at symptom onset compared to patients with OTs (MWU = 9810.5, *p* < 0.001). 

#### 3.3.2. Patient Comorbidity

Despite the group differences seen in the descriptive statistics in Table 2, the regression demonstrated no relationship between the patients’ psychiatric comorbidities and their final diagnosis (FTLB or OT) after correcting for age and sex. 

#### 3.3.3. Family History

The diagnoses of the patients’ first-degree relatives differed significantly between the two groups. Significantly more patients with FTLB had relatives with other psychiatric disorders (χ^2^(1) = 13.008, *p* < 0.001), whereas significantly more patients with OTs had relatives with tics (χ^2^(1) = 9.67, *p* < 0.001)

#### 3.3.4. Potential Triggers

The presence of a psychosocial trigger (*p* < 0.001) was significantly associated with a diagnosis of FTLB after correcting for sex and age. Patients with FTLB had significantly higher odds of experiencing a psychosocial trigger immediately before their tic onset (OR = 59.620, 95% CI (16.6, 213.3)).

#### 3.3.5. Tics Type

The presence of simple motor tics (*p* < 0.001) was significantly associated with patient diagnosis. Patients with FTLB had significantly reduced odds of experiencing any simple tics (OR = 0.14, 95% CI (0.001, 0.142)). However, patients with FTLB had increased odds of experiencing self-injurious behaviors (OR = 41.23, 95% CI (3.96, 429.7), *p* = 0.002), unrestrained speech (OR = 18.542, 95% CI (1.644, 209.099), *p* = 0.018) and coprophenomena (OR = 14.963, 95% CI (1.719, 130.226), *p* = 0.014)

#### 3.3.6. Tics Number

FTLB patients had significantly fewer simple motor tics (*p* < 0.001, OR = 0.213, 95% CI (0.11, 0.41)) but significantly more complex motor tics (*p* < 0.01, OR = 2.91, 95% CI (1.52, 5.582)). 

## 4. Discussion

The purpose of this article was to examine the previously shown and suggested differences between patients with FTLB and patients with OTs. A series of analyses revealed that the patients with FTLB were significantly more likely to be female, had an increased age at tic onset and were significantly more likely to have first-degree relatives with other psychiatric diagnoses. Furthermore, the FTLB patients had higher odds of experiencing self-injurious behaviors, unrestrained speech, and coprophenomona and had significantly more complex tics. Conversely, the FTLB patients had fewer first-degree relatives with tics, and reduced odds of both presence and high number of simple motor tics. Thus, this study was able to locate multiple differences between patients with FTLB and OTs in the Danish Cohort (Table 6). 

### 4.1. Differences in Family History

The results revealed a significant difference in the patient groups’ family histories as significantly more patients with FTLB had relatives with psychiatric disorders and significantly fewer had relatives with tics compared to patients with OTs. FTLB patients’ family history of tics is an area with conflicting results. Decreased prevalence of tics has been reported previously [6,31], but other studies have found no difference in family history between OT and FTLB patients [28,30]. In their diagnostic checklist, Trau and colleagues propose using the lack of a family history of tics as one of the criteria for FTLB [31]. While the results from this study are in line with their report, multiple papers show no difference in family history; thus, including family history as a diagnostic criterion should be approached with caution. Pringsheim and colleagues (2023) also consider these conflicting results, and family history is not included on their list of diagnostic criteria [32].

This paper is the first to show an increased likelihood of neuropsychiatric diagnoses when assessing the family history of patients with FTLB compared to that of patients with OTs. The only other paper to examine this potential point of interest is Han and colleagues (2022), which found no differences [30]. This discrepancy in results may be due to our decision to group all psychiatric disorders, except ADHD, ASD, and anxiety, into one category (other psychiatric disorders) thereby increasing the sample size.

The increased prevalence of psychiatric disorders in the family indicates that having first-degree relatives with psychiatric disorders may significantly increase the risk of developing FTLB. There are two possible mechanisms that could underlie this increased risk. Like OTs, which are assumed to result from genetic transmission of an underlying neurochemical imbalance [11], FTLB patients may inherit a psychiatric vulnerability that predisposes them to develop functional disorders. For example, it has been shown that mothers who are severely stressed during the pregnancy or have themselves experienced child abuse are more likely to have children with a dysregulated stress response system, which may lead to an increased risk of the children developing a functional disorder [37]. Alternatively, having first-degree relatives with a psychiatric disorder may lead to more frequent experiences of psychosocial triggers and a lack of resources within the family to help the child with dealing with adverse events, leading to an increased risk of developing FTLB. To determine which of these options reflects the truth most closely, more research is required.

These results also point to an important clinical perspective. When treating pediatric patients with FTLB it may be more important than in other patient groups to consider the resources and stability of the family unit. If one of the first-degree relatives does have a psychiatric disorder, the family may need more assistance to support the child’s treatment.

### 4.2. Differences in Triggers

A result unique to this paper is the significantly higher prevalence of a trigger before or at symptom onset in patients with FTLB compared to patients with OTs. Significantly more FTLB patients had experienced a psychosocial adverse event, such as bullying or family conflict immediately before the onset of their symptoms. This finding corroborates with observations made by other authors, who note the presence of a trigger in many patients with FTLB [2,6,27]. The presence of a trigger is not included in Trau and colleagues’ diagnostic criteria but is included as a part of a minor criteria in Pringsheim and colleagues’ diagnostic checklist [31,32]. Considering the uniformity of the literature it may be worth considering whether trigger presence should be added as a more prominent criterion. However, it should be mentioned that there were patients with OTs who did report a trigger, and patients with FTLB who did not, so it should not be a deciding factor.

The difference in trigger prevalence between patients with FTLB and OTs found in this study may be due to the differences in the etiology of the disorders. While OTs are thought to develop mostly without input from outside forces [38] many patients with FMDs are believed to develop the disorder in direct response to adverse events [1,39]. The suspected mechanism is that the patients experience a detrimental sensory input, such as the physiological discomfort of stress and anxiety, and due to a mix of risk factors, such as excessive rumination or disordered illness beliefs, interpret these sensory inputs as a sign of an illness, for example, tics. Believing they have the disorder, the patients expect to develop the symptoms, and this prediction becomes so strong that symptoms actually do occur [3,39]. The results of this study support this theory for FTLB by showing a significantly higher trigger prevalence compared to the OT group. Furthermore, this theory indicates that inquiring about potential adverse events and the patients’ response to these may not only be important for differential diagnostics. It may also enable clinicians to provide guidance to patients about illness beliefs and anxious ruminations, and as such prevent them from potentially developing new functional symptoms as a response to the next adverse event.

### 4.3. Tic Differences

This study revealed that patients with FTLB had an increased risk of developing coprophenomena, self-injurious behaviors, and unrestrained speech compared to patients with OTs. This is in line with other studies and observations [27,30,31]. The presence of these behaviors may help differentiate patients with FTLB from patients with OTs, and it is included on both lists of diagnostic criteria [31,32].

Apart from a higher rate of complex tics, the patients with FTLB were also less likely to have simple tics and had fewer simple tics compared to patients with OTs. While this lack of simple tics in patients with FTLB has previously been noted in the literature [27,28], neither Pringsheim and colleagues nor Trau and colleagues include it as a diagnostic criterion, instead focusing on the patient group’s complex tics [31,32]. However, this reduction in simple tics could present a way to reliably differentiate FTLB patients from female OT patients. A growing concern within the literature has been the misdiagnosis of especially female TS patients as having FTLB with reference to these patients’ later tic onset and higher number of complex tics which worsen over time and tend to have a higher functional impact [7,33,40]. Conversely, studies comparing the prevalence of simple tics between female and male OT patients do not find a difference [41,42]. Thus, focusing on a lack of simple tics, rather than a high level of complex tics may be a way to differentiate female TS and FTLB patients.

### 4.4. Lack of Impact of Comorbidities

One of the surprising results from this study is the lack of differences in comorbidities between patients with FTLB and OT patients. While comorbidities and other symptoms will always vary between patients, it is generally accepted that anxiety and depression are more prevalent comorbidities in patients with FTLB while OCD and ADHD are more prevalent in OT patients [2,4]. Anxiety and depression as comorbidities are also included as a minor criterion in Pringsheim and colleagues’ checklist and as a major criterion in Trau and colleagues’ checklist [31,32]. However, the literature comparing the comorbidity profiles of OT and FTLB patients is inconsistent. Trau and colleagues (2022) and Han and colleagues (2022) both find a higher prevalence of anxiety and depression among their FTLB sample compared to their OT sample [30,31]. However, neither of these papers corrects for age or sex which is particularly important when considering that anxiety and depression are known to be more prevalent in women [43,44], and that the increased age of FTLB patients means they have had comparatively longer time to receive a diagnosis. The article by Pringsheim and colleagues (2021) [29] which does correct for age and sex finds a higher prevalence of depression and anxiety in their FTLB group, while Paulus and colleagues (2022) [28] which compares age- and sex-matched OT and FTLB patients does not. A reduction in the prevalence of ADHD and OCD diagnosis in FTLB patients compared to OT patients has, to our knowledge, not been established in any study comparing the two patient groups after correcting for age and sex. Thus, there may not be any substantial differences in comorbidity profiles between the patient groups. While comorbidities are included in both diagnostic checklists [31,32], these conflicting results indicate that emphasis should be placed on other criteria.

### 4.5. Limitations

This study has several limitations. First, to enable a clear comparison between patients with FTLB and OTs it was decided to exclude all patients with both FTLB and OTs. This exclusion means that the interaction and overlap between FTLB and OTs cannot be examined here, but it should be explored elsewhere to further clarify the differential diagnosis. The second limitation is that the data was not initially collected using a standardized, scripted research procedure. While all the information used in this study, such as family history or comorbidities, is a standard part of the systematic medical interview conducted in the clinic, the lack of a standardized research procedure could have resulted in differences in how the questions were asked and what details were of note to the attending medical personnel. This might be particularly important for the results concerning psychological triggers as the questions might have introduced recollection bias depending on how they were asked. Another limitation is that this study is a retrospective study using medical records meaning that not all relevant information was collected. For example, the presence of depression was not recorded individually for the patients, and no YGTSS scores were available for the FTLB patients, as these patients are not scored at the clinic. This is, however, a limitation of all retrospective studies. As in all other current studies of FTLB, the diagnosis was made based on the opinion of an experienced child neurologist rather than standardized criteria, which does increase the risk of circular reasoning. For example, are patients with FTLB older and more likely to be female compared to patients with OTs, or are clinicians just more likely to diagnose the tics of teenage girls as functional [4]? Furthermore, there is a chance that the phenomenology of FTLB may change, as the trends on social media do.

## 5. Conclusions

During the COVID-19 pandemic, an increasing number of patients with FTLB were seen worldwide. This development has made it increasingly important to localize reliable points of differentiation between these patients and patients with OTs that may be included in an FTLB diagnosis. This paper presents one of the largest studies on FTLB yet comparing patients with FTLB and OTs on a wide range of possible differentiation points. The results show that patients with FTLB had more complex tics and were more likely to be female, to have family members with psychiatric disorders, and to have experienced triggers before the onset of their symptoms, which arrived significantly later. Conversely, the FTLB patients had significantly fewer simple tics and were less likely to have family members with OTs. These results support the inclusion of complex tics in diagnostic checklists for FTLB, as seen in the checklists by both Pringsheim and colleagues and Trau and colleagues [31,32]. Furthermore, it may be beneficial to include a lack of simple tics as a key point in these checklists as well, as the lack of simple tics could help differentiate patients with FTLB from female patients with OTs. Conversely, this study found no support for patient comorbidity as a point of differentiation between the two disorders. Comorbid anxiety and depression are included as criteria supporting the FTLB diagnosis in both checklists [31,32], however, the literature on this topic is conflicted. As such, it may be inadvisable to emphasize patient comorbidity in the diagnostic process. Despite the increasing amount of literature on FTLB, there is still much to be discovered. The trajectory of the disorder is virtually unknown, as only a few follow-up studies have been published [45,46]. Furthermore, the etiology of FTLB and functional disorders in general is still heavily discussed, although the significantly increased presence of a psychosocial trigger in the FTLB population does support adverse events as one possible cause. Future research may explore these topics further.

## Figures and Tables

**Table 1 healthcare-11-01481-t001:** Overview of all included variables used in the analyses, along with the confounders. ADHD = Attention Deficit Hyperactivity Disorder, OCD = Obsessive Compulsive Disorder, ASD = Autism Spectrum Disorder.

Analyses	Outcome Variable	Potential Confounders Included in the Analysis	Type of Analysis
Sex	Sex		Chi-Square test
Age at onset	Age at onset		Mann-Whitney U
Patient comorbidity	Presence of anxiety, ADHD, OCD, ASD, and other psychiatric diagnoses	Sex and age at first visit	Binary logistic regression
Family history	Presence of anxiety, ADHD, ASD, tics and other psychiatric diagnoses in first-degree relatives		Chi-Square test
Potential Triggers	Presence of psychosocial triggers	Sex and age at first visit	Binary logistic regression
Tics type	Presence of simple motor tics, complex motor tics, simple vocal tics, and complex vocal tics	Sex and age at first visit	Binary logistic regression
Number of tics	Number of simple motor tics, complex motor tics, simple vocal tics, and complex vocal tics	Sex and age at first visit	Negative binomial regression
Specific behaviors	Presence of self-injurious behaviors, long bursts of tics, unrestrained speech, and coprophenomena	Sex and age at first visit	Binary logistic regression

**Table 2 healthcare-11-01481-t002:** Descriptive statistics showing the number and percentage of patients with respectively FTLB and OTs who were diagnosed with different psychiatric comorbidities before their referral and had experienced triggers before their symptom onset. FTLB = Functional Tics Like Behavior, OTs = Organic Tics, ADHD = Attention Deficit Hyperactivity Disorder, OCD = Obsessive Compulsive Disorder, ASD = Autism Spectrum Disorder.

	FTLB (n = 53)	OTs (n = 200)
Number of patients with ADHD	10 (18.9%)	31 (15.5%)
Number of patients with anxiety	14 (26.4%)	13 (6.5%)
Number of patients with ASD	7 (13.2%)	15 (7.5%)
Number of patients with OCD	10 (18.9%)	16 (8%)
Number of patients with other psychiatric diagnoses	15 (28.3%)	12 (6%)
Number of patients with a psychosocial trigger before/at symptom onset	42 (79.2%)	27 (13.5%)

**Table 3 healthcare-11-01481-t003:** Descriptive statistics showing the number and percentage of patients with respectively FTLB and OTs who had first-degree relatives with different psychiatric diagnoses. FTLB = Functional Tics Like Behavior, OTs = Organic Tics, ADHD = Attention Deficit Hyperactivity Disorder, ASD = Autism Spectrum Disorder.

	FTLB (n = 53)	OTs (n = 200)
Number of patients with first-degree relatives with ADHD	7 (13.2%)	25 (12.5%)
Number of patients with first-degree relatives with anxiety	9 (17%)	19 (9.5%)
Number of patients with first-degree relatives with tics	6 (11.3%)	66 (33%)
Number of patients with first-degree relatives with ASD	5 (9.4%)	11 (5.5%)
Number of patients with first-degree relatives with other psychiatric diagnoses	16 (30.2%)	21 (10.5%)

**Table 4 healthcare-11-01481-t004:** Descriptive statistics showing the number and percentage of patients with respectively FTLB and OTs who had different types of tics. FTLB = Functional Tics Like Behavior, OTs = Organic Tics.

	FTLB (n = 53)	OTs (n = 87)
Number of patients with simple motor tics	39 (73.6%)	83 (95.4%)
Number of patients with complex motor tics	44 (83%)	48 (55.2%)
Number of patients with simple vocal tics	30 (56.6%)	49 (56.3%)
Number of patients with complex vocal tics	30 (56.6%)	13 (14.9%)
Number of patients with self-injurious behaviors	23 (43.4%)	4 (4.6%)
Number of patients with unrestrained speech	25 (47.2%)	1 (1.1%)
Number of patients who have experienced long bursts of tics	7 (13.2%)	3 (3.4%)
Number of patients with coprophenomena	16 (30.2%)	2 (2.3%)

**Table 5 healthcare-11-01481-t005:** Descriptive statistics showing the median number of different types of tics experienced by respectively patients with FTLB and patients with OTs. FTLB = Functional Tics Like Behavior, OTs = Organic Tics.

	FTLB (n = 53)		OTs (n = 87)	
Number of Tics	Median	Range	Median	Range
Simple motor tics	1	0–10	4	0–10
Complex motor tics	3	0–10	1	0–6
Simple vocal tics	1	0–5	1	0–2
Complex vocal tics	1	0–14	0	0–2

**Table 6 healthcare-11-01481-t006:** Overview of the results of the study. Underlined results are results replicated from other studies, while bold font indicates new findings from this study. Finally, italics indicate results from other studies which could not be replicated in this one. The other studies which found each result are listed in the “references” column. FTLB = Functional Tics Like Behavior, OTs = Organic Tics.

FTLB Patients Compared to OT Patients	References
Older at onset	[28,29,30,31] ^1^
More likely to be female	[29,30,31] ^2^
Decreased likelihood of tics in the family	[31]
More complex motor tics after correcting for age and sex	[28]
**Increased likelihood of a psychiatric diagnosis in the family**	
**Less likely to have simple motor tics, and fewer simple motor tics in total after correcting for age and sex**	
**More likely to have self-injurious behaviors, coprophenomena and unrestrained speech after correcting for age and sex**	
**Increased likelihood of psychosocial trigger before onset after correcting for age and sex**	
*Increased likelihood of anxiety and other psychiatric diagnoses as comorbidities*	[29,30,31]

^1^ Mean age at onset across studies: FTLB (n = 95): 13.8–16.5 years, TS (n = 559): 5–6.8 years ^2^ Sex ratio across studies: FTLB (n = 82): 61–100% female, TS (n = 546): 21–61% female.

## Data Availability

The data presented in this study are available on request from the corresponding author. The data are not publicly available due to patient confidentiality.

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
