# Peer review of "Clarifying the Differences between Patients with Organic Tics and Functional Tic-Like Behaviors"

_healthcare, 2023, doi:10.3390/healthcare11101481_

Round 1
Reviewer 1 Report
Thank you so much for giving me the opportunity to review this important article. The topic investigated in this study is relevant and the data presented are important, I also congratulate the authors on impressive number of patients with FLTB, I believe this is the biggest sample published so far or one of the biggest. However, I have some suggestions that the authors should consider to introduce prior to publication. First of all, I would suggest to consult an English proofing service or the help of native speaker since many terms used in the text are hard to understand. For example, I would suggest to change the word "debut" to "onset" in the whole manuscript. Also keywords are not clear for me, what is diagnostic medicine? I would suggest to change it. Also Tourette syndrome should be spelled with capital letters. Also, I would suggest to ellaborate more on the findings in the abstract since now they are too vague. FLTB corresponds to functional tic like behaviors and not behavior, please correct. I would also suggest to change the term FMD for functional movement disorders and not functional motor disorder or functional neurological disorder (FND). When talking about typical phenomenology of organic tics, I would suggest to also include recently published paper by Nilles et al. (2023): Nilles C, Martino D, Fletcher J, Pringsheim T. Have We Forgotten What Tics Are? A Re-Exploration of Tic Phenomenology in Youth with Primary Tics. Mov Disord Clin Pract [Internet]. 2023 [cited 2023 Apr 1]; Available from: https://onlinelibrary.wiley.com/doi/full/10.1002/mdc3.13703. When talking about previous papers about FTLB, the authors missed some contributions, please incorporate this paper: Janik P, Milanowski L, Szejko N. Psychogenic tics: Clinical characteristics and prevalence. [Internet]. Vol. 48, Psychiatria Polska. 2014. p. 835–45. Similarly, although I agree that one of the denominations used to talk about recent surge of FTLB is "TikTok Tics", there are also other terms proposed by the scholars worldwide so I would suggest to include them as well as appropriate citations. Similarly, not all papers on this topic have been included such as: Buts S, Duncan M, Owen T, Martino D, Pringsheim T, Byrne S, et al. Paediatric tic-like presentations during the COVID-19 pandemic. Arch Dis Child [Internet]. 2022 Mar 1 [cited 2023 Jan 25];107(3):e17. Available from: https://pubmed.ncbi.nlm.nih.gov/34824091/; Zea Vera A, Bruce A, Garris J, Tochen L, Bhatia P, Lehman RK, et al. The Phenomenology of Tics and Tic-Like Behavior in TikTok. Pediatr Neurol [Internet]. 2022 May 1 [cited 2023 Jan 25];130:14–20. Available from: https://pubmed.ncbi.nlm.nih.gov/35303587/; Fremer C, Szejko N, Pisarenko A, Haas M, Laudenbach L, Wegener C, et al. Mass social media-induced illness presenting with Tourette-like behavior. Front Psychiatry [Internet]. 2022 Sep 20 [cited 2022 Nov 27];13. Available from: https://pubmed.ncbi.nlm.nih.gov/36203825/; Martino D, Hedderly T, Murphy T, Müller-Vahl KR, Dale RC, Gilbert DL, et al. The spectrum of functional tic-like behaviours: Data from an international registry. Eur J Neurol [Internet]. 2023 Feb 1 [cited 2023 Mar 1];30(2):334–43. Available from: https://pubmed.ncbi.nlm.nih.gov/36282623/; Hull M, Parnes M. Tics and TikTok: Functional Tics Spread Through Social Media. Mov Disord Clin Pract [Internet]. 2021 Nov 1 [cited 2023 Mar 7];8(8):1248–52. Available from: https://pubmed.ncbi.nlm.nih.gov/34765689/; Olvera C, Stebbins GT, Goetz CG, Kompoliti K. TikTok Tics: A Pandemic Within a Pandemic. Mov Disord Clin Pract [Internet]. 2021 Nov 1 [cited 2023 Mar 7];8(8):1200–5. Available from: https://pubmed.ncbi.nlm.nih.gov/34765687/; Nagy P, Cserháti H, Rosdy B, Bodó T, Hegyi M, Szamosújvári J, et al. TikTok and tics: the possible role of social media in the exacerbation of tics during the COVID lockdown. Ideggyogy Sz [Internet]. 2022 [cited 2023 Mar 7];75(5–06):211–6. Available from: https://pubmed.ncbi.nlm.nih.gov/35819338/;Okkels KB, Skov L, Klanso S, Aaslet L, Grejsen J, Reenberg A, et al. Increased Number of Functional Tics Seen in Danish Adolescents during the COVID-19 Pandemic. Neuropediatrics [Internet]. 2023 [cited 2023 Jan 26]; Available from: https://pubmed.ncbi.nlm.nih.gov/36417931/; when talking about methodology, I am not sure why the authors say it was a journal review, it is confusing since this is not a review, but original study, I would suggest correcting this aspect. The authors should also mention why only the period starting at 2020 was included in the analysis? We know that this phenomenon was present also before, so the authors should elaborate on this criterion. I have also suggestion regarding the comparison of tic repertoire in FLTB and OT, I would suggest to compare YGTSS in both groups, but I am not sure whether the authors have this information? I am also not sure why the authors picked these types of tics as a separate category: “self-harming tics, tic attacks, uncontrolled speech and coprophenomena”. I agree that self-injurious behaviors (SIB) should be recorded as separated item, but more appropriate term would be SIB, since these behaviors are not always secondary to tics, but could also be a manifestation of psychiatric comorbidities. The authors should elaborate on the term tic attacks –this is only for FLTB group, correct? This is highly atypical for OT. I agree that coprophenomena should be mentioned as separate category, but why uncontrolled speech? Instead, more important are complex movements and vocalizations, especially such as throwing objects, since it is highly atypical for OT. I am also not sure whether the name “extreme tics” is appropriate since also other phenomena could be included in this category and, on the contrary, SIB and uncontrolled speech could be part of tics or psychiatric comorbidities. Please correct the description for Table 1 as follows: “Overview of all included variables used in the analyses, along with the confounders”. Please also correct this sentence since it is hard to understand “These patients did not differ from the 23 FTLB patients who had not consumed tics-centered social media, except for having higher odds of self-injurious behaviors (OR = 5.061, 95% CI [1.37, 18.72], p = 0.015)”. Please include in Tables 2-6 also p value along with number of patients with particular phenomenon. The information from Table 4 should be included in some of the other Tables since it includes only one variable. For Table 6 – range should be at least two values from minimum to maximum, and in now it has only one, could you please correct it? Please also correct the section about number of simple/complex tics as follows, since now it is hard to understand: “While number of simple motor tics was significantly higher in OT group (p < 0.001), number of complex motor tics was significantly higher in the group of FLTB (p = 0.01).”. Please include also odds ratio in this sentence and not as separate sentence. Please also correct the paragraph about “extreme tics” depending on the corrections from the methods. For Table 7, I really like the idea, but I would suggest to correct it since now it is too confusing and not clear, I would suggest to insert references to the studies that found the same findings, unique findings, and also include more details, for example sample size, age, sex, age of onset. Please correct in the limitations that this was a journal study, because it is confusing.
Reviewer 2 Report
1. General comments
This study examines the differences between Functional Tic-like Behaviors (FTLBs) and Organic Tics (OTs) in a Danish cohort by analyzing factors such as family history, triggers, tic characteristics, and comorbidities. The study included 53 FTLB patients who visited The National Tourette's Syndrome Clinic from May 2020 to April 2022 and 200 OT patients who attended the clinic from May 2020 to July 2021. By investigating the largest pediatric FTLB patient population and comparing them to pediatric OT patients, this study provides unique insights into the disparities between these groups and potential clinical implications. Some suggestions are offered to enhance the clarity of the manuscript.
2. Specific comments
a) Major
i) No major comments.
b) Minor
i) On page 5, line 202, the statement " Compared to the 200 patients with OTs, a higher percentage of the patients with FTLBs had psychiatric comorbidities (Table 2)" should include the overall percentages and the results of the chi-squared test for a more accurate presentation. For example: "Compared with the 200 patients with OTs, a higher percentage of patients with FTLBs had psychiatric comorbidities (XX% in FTLBs, XX% in OTs, [results of the chi-squared test]) (Table 2)".
Reviewer 3 Report
- SUMMARY: This paper aims to expand our understanding of the profile of patients presenting with organic tics as compared to functional tic-like movements. It is an augmentation to a prior paper by the same authors. (See Okells KB, et al.Increased Number of Functional Tics Seen in Danish Adolescents during the Covid-19 Pandemic. Neuropediatrics. PUblished online January 12, 2023.) This paper adds to that prior work in several ways. First, it assesses a larger cohort of patients with FTLB. Second, it compares the group of patients with FTLM to a control group of patients with with OT. Finally, and most importantly, the authors corrected for age and sex in the analysis of psychiatric comorbidities in these two groups, and showed no difference when this adjustment was made. This is a novel finding.
- General Comments: FTLMs were very common during the height of the pandemic. While the incidence of FTLM is decreasing, I still think there is a lot to learn from this event. Overall, the design of this study is straight-forward, and the findings are helpful for the clinical community. The methods for how triggers were assessed were not clearly described, and I worry there may be bias in how a history of triggers was elucidated in clinical interviews, leading to recall bias. Furthermore, while corrections were made for age and gender when analyzing co-occuring neuropsychiatric conditions, this was not done for triggers. Given the differences in age and gender between the OT and FTLM groups, such a correction would be important in assessing the validity of the authors' findings regarding triggers in FTLM. Finally, different ascertainment methods were used in assessing tic severity and number for patients with FTLM and OT. I assume this was because YGTSS's were not obtained standardly for patients with FTLM. This is a limitation of a retrospective study, but may have skewed findings related to the presence of absence of simple tics. I don't think either of these things should limit publishability after modifications, but they must be acknowledged.
Line 70 – I don’t think the word “recently” can still be applied here. The epidemic of FTLM has been discussed widely both in the media and academic publications, and the relative incidence of FTLM have steadily declined over the past year.
Line 111-112: The other way to get around this would be a large registry combining data from multiple sites. Thus you cannot say that repeated replication is the ONLY way to address this problem. I understand how this applies to justification of this study. It may be more correct to say “impairment in generalizability can be ameliorated by replication from different research groups.”
112-119: I think this is an important argument for what your paper adds to the scientific literature. I found these lines to be a little clunky. It would be helpful to further clarify so this novel finding shines throughout the paper.
Line 135: Consider changing the word "on" to "via" or "in".
Section 2.2.3: I would like more information on what the stressors were (breakdown of categories). Also, it would be helpful to know how this information was obtained. For example, did the author's approach the historical interview the same way when FND was suspected as compared to OTD?
Lines 158-161: How were these questions asked? Was there a script, or were they asked as part of the typical interview process? I worry, if the latter, that there would be bias in HOW these questions were elicited that would lead to recall bias. For example, if we are truthful, MOST patient in this trial should have said “yes” to deterioration in mental and/or wellbeing due to changes surrounding the pandemic. Finally, you said that these things were recorded in a yes/no approach. Were different types of stressors recorded? Were there certain types of stressors that were more prominent?
Line 167: Why didn’t you use the YGTSS for both?
Line 169: Take out the word “Here.”
Line 173: I was a little confused when you said “these extreme tics were both registered separately.” Are you here referring to vocal and motor coprophenomenon? Did you count each coprophenomenon as a tic type or just mark the category of “coprophenomenon.” I am not seeing a place where they were addressed separately in your outcomes.
Table 1: I wish you had separated depression from other psychiatric diagnoses. Depression itself is over-represented in both FND and TS, and has a very different profile than thought disorders, bipolar disorder etc. If you have this data available, it may be worth including what percentage of patients with "other psychiatric diagnosis" had depression versus another disorder.
Section 3.1: Much of this could be moved to a chart OR moved to section 3.2. I would consider moving the entirety of lines 202-216 to the respective subsets in section 3.2. This will allow you to highlight your findings in a more thoughtful way. It would also allow you to combine line 202-203 with section 3.2.2. Having them seperate was initially confusing to me.
Finally, it may be interesting here to include how many patients were excluded due to the co-occurrence of FTLM and OT. That in and of itself would add something new to the literature.
Section 3.2.4. Again, it would be helpful to have more information here. In addition to having more information on how this information was obtained, I would like to see what kinds of psychosocial stressors were experienced. I also feel it would be helpful to correct by age and gender in this section as those two factors may affect the frequency of stressors.
Table 7: Nothing was bolded or underlined in the copy of the article that I was provided.
Section 4.1: You make several important points in this section. In consideration of the interplay between genetic risks and environmental risks from having a family member affected by a mental health disorder, it may be worthwhile to consider the role of epigenetics in FND. (See Frodl T. Do (epi) genetics impact the brain in functional neurological disorders? Handb Clinc Neurol (2016) 139: 157-65.)
Line 302: Consider changing psychiatric to biological or neuropsychiatric.
Line 332: There is a random number 4 after the word tic. I am assuming this is a typo.
Section 4.2. I would still love more info on what the triggers were and whether this finding holds true after correction for age and gender.
Line 345: Consider changing the phrase "thus be a promising way to" to "help".
Line 348: Consider taking out the word "also."
Line 362-363: I would be careful here. Neuroticism and perfectionism, while not the same as OCD, are common personality traits in FND patients. Furthermore, depression and anxiety are also high in children with TS, and depression/anxiety scales may be even higher in studies of adults with TS.
Section 4.5: Additional limitations worth considering include:
1. The fact that the YGTSS was used only for OT’s, not FTLBs. The reason for this was likely because this WAS a retrospective study, which is fine, but it may impact the recall of things like the presence simple motor tics in the FTLB group that might not be mentioned if not specifically asked about.
2. Recall bias regarding triggering events depending on how the questions were asked.
Reviewer 4 Report
The authors conducted a retrospective case-control study investigating the difference between Functional Tic-Like Behavior (FTLB) and organic tics (OT) in comorbidities, familial history, and symptom characteristics. The authors found that FTLB patients were likelier to be female and have psychiatric comorbidities, less likely to have familial history of tic, etc. The difference between FTLB and OT is scarcely investigated so far, therefore I believe this study is worthwhile to neurologists or psychiatrists.
Although this is a retrospective study, it is conducted in a faculty specialized for Tic, and the patient information is described in detail. However, as the authors stated in the limitation section, the characteristics of FTLB listed in Table 7 may be a result of circular reasoning. The senior child neurologists may have used the information listed in Table 7 to diagnose the patients with FTLB or OC, and thus these differences may be found. I suggest the authors describe briefly how they differentiated these two entities. And also, clarifying the content of the systematic medical interview at the first clinic visit will help the readers' interpretation because it was the fundamental method to obtain the data used in this study.
Minor concern; in Table 7, the authors state that "italics indicates results from other studies which could not be replicated in this one." The item "Increased likelihood of anxiety and other psychiatric diagnoses as comorbidities" is written in italics and thus the authors think this finding is not replicated in their study, However, from Table 2, it seems that this finding IS replicated in their study. (For example, the number of patients with anxiety is 26.4% in FTLB, which is much higher than 6.5% in OT). I think this is an inconsistency between Table 2 and Table 7 and thus should be corrected appropriately.
Round 2
Reviewer 1 Report
Thank you so much for introducing changes proposed by me. I think that the paper improved significantly and recommend it for publication.